# Patient Involvement in Shared Decision-Making: Do Patients Rate Physicians and Nurses Differently?

**DOI:** 10.3390/ijerph192114229

**Published:** 2022-10-31

**Authors:** Maura Galletta, Maria Francesca Piazza, Stefania Luisa Meloni, Elsa Chessa, Ilenia Piras, Judith E. Arnetz, Ernesto D’Aloja

**Affiliations:** 1Department of Medical Sciences and Public Health, University of Cagliari, 09100 Cagliari, Italy; 2Liguria Health Authority (A.Li.Sa.), 16121 Genoa, Italy; 3Intensive Care Unit, Binaghi Hospital, ASL Cagliari, 09100 Cagliari, Italy; 4Emergency Department, SS. Trinità Hospital, ASL Cagliari, 09100 Cagliari, Italy; 5Department of Family Medicine, Michigan State University, Grand Rapids, MI 48824, USA

**Keywords:** shared decision-making, nurses, physicians, patient involvement, patient satisfaction, perceived quality of care

## Abstract

Background. Shared decision-making implies that patients and healthcare professionals make decisions together about clinical exams, available treatments, choice of options, and the benefit or downside of every choice. Patients involved in the shared decision-making process are more compliant with treatments and have a reduced risk of complications related to the pathology. In Italy, patient involvement in caring processes is still barely investigated. Aim. To investigate patients’ perceptions about shared decision-making with physicians and nurses, respectively, and to examine the relationship between shared decision-making and patient satisfaction and perceived quality of care/treatment. Methods. The study was performed between March and June 2019 in two wards of one Italian hospital. A questionnaire was administered to inpatients at the time of admission and again at discharge, including demographic information and measurement scales regarding patient involvement in shared decision-making, patient satisfaction, and perceived quality of treatment/care. Results. A total of 151 out of 301 patients completed questionnaires at both admission and discharge. Patients’ scores for shared decision-making (information, patient needs, treatment planning) were significantly different for physicians and nurses. At both admission and discharge, patients rated shared decision-making significantly higher for physicians compared to nurses, while there were no differences in their satisfaction ratings. Patient ratings of physicians did not change from admission (information: mean (M) = 3.50, standard deviation (SD) = 0.81; patient need: M = 3.05, SD = 1.05; treatment planning: M = 2.75, SD = 1.23) to discharge (information: M = 3.50, SD = 0.79; patient need: M = 3.17, SD = 1.02; treatment planning: M = 2.66, SD = 1.23) (*p* = 0.924, *p* = 0.098, *p* = 0.293, respectively), but patients’ ratings of nurses’ behavior increased significantly from admission (information: M = 2.44, SD = 1.23; patient need: M = 2.27, SD = 1.17; treatment planning: M = 2.12, SD = 1.19) to discharge (information: M = 2.62, SD = 1.22; patient need: M = 2.53, SD = 1.24; treatment planning: M = 2.35, SD = 1.21) (*p* = 0.019, *p* = 0.001, *p* = 0.003, respectively). Attention to patients’ needs was the key determinant of both satisfaction with nurses (OR = 3.65, 95% CI = 1.31–10.14, *p* = 0.013) and perceived quality of care (OR = 3.97, 95% CI = 1.49–10.55, *p* = 0.006). Providing appropriate information about disease progress and treatments was a key determinant of both satisfaction with physicians (OR = 19.75, 95% CI = 7.29–53.55, *p* < 0.001) and perceived quality of treatment (OR = 8.03, 95% CI = 3.25–19.81, *p* < 0.001). Discussion. Nurses should be sensitized to involving patients in the decision-making process, especially upon hospital admission. Specific training about effective communication techniques can be implemented to manage relationships with patients in different caring situations. Practical implications and future directions are discussed.

## 1. Introduction

In 1973, the American Hospital Association approved the Patient’s Bill of Rights, which preserves the patient’s right to be informed and to be involved in therapeutic decisions. To date, the model that more represents this aim is the shared decision-making (SDM) model. At the base of the model, there is the respect for the principle of individual self-determination [1] as a desirable goal for patients, and professionals should strive to achieve this goal. In this model, patients and healthcare professionals (physicians, nurses, etc.) make decisions together, using the best available evidence [2]. Patients are encouraged to think about their clinical exams, available treatments, choice of options, and the benefits or downsides of every choice in order to communicate their preferences and help healthcare professionals in the decision-making process. The SDM model respects patients’ autonomy and promotes their participation by building good relationships and respecting both individual competence and interdependence on others [3]. These ethical principles extend the concept of informed consent as something that goes beyond the mere transfer of information but includes respect for informed preferences [4].

Several studies have highlighted the importance of a patient’s active participation in their care [5,6]. It is considered a way to increase awareness of one’s own health condition, which raises motivation to take responsibility and a sense of control for the treatment [7]. Patient participation also positively affects behavior change (e.g., lifestyle) and adherence to prevention strategies to reduce the risk of problems or complications related to the pathology [8,9,10]. In particular, some authors highlighted that promoting shared decision-making and defining shared goals between healthcare professionals and patients improved adherence to healthy behaviors and outcomes, especially in patients with pre-existing underlying chronic conditions [11,12]. Involving patients in the shared decision-making process and incorporating patients’ preferences into the process helps patients make decisions, thereby improving their adherence to prevention strategies [7].

Moreover, active participation in care processes can reduce the risk of clinical errors; patients can provide information about their clinical history to healthcare professionals and communicate possible drug collateral effects. They can also report difficulties with continuing care or can recognize medical oversights, thus reducing the risk of adverse events [6]. The additional value of this interactive approach between patient and care provider is increased patient autonomy and empowerment, involving the sharing of ideas, perspectives, and intentions in the caring process [13,14].

Patient involvement is necessary for shared decision-making and occurs when patients learn to take responsibility for their own health and solve problems with information and support from professionals. Patient empowerment begins with a healthcare provider who recognizes that patients are in control of their care and aims to increase the patient’s ability to think critically and make autonomous and informed decisions about their own health [6,9].

Literature shows that patients actively involved in diagnostic and therapeutic decisions achieve better treatment results [15]. According to a pioneer study [16], when nurses introduce themselves to the patient and explain what they will do, patients perceive that nurses have a sense of control over the situation. Moreover, patients desire to express their ideas and perceive that their opinions and/or concerns are received and listened to by nurses.

According to a systematic review [17], involving patients in decision-making regarding their treatment helps to improve patients’ knowledge about available options, clearing possible risks and benefits of the treatment, and making diagnostic choices that are consistent with their preferences and values. In this sense, patient involvement increases patient participation in the whole decision-making process [17]. Concerning diagnostic choices, the SDM model shows that informing patients about the positive and negative aspects of medical exams and various treatment choices can improve the safety and quality of diagnostic interventions [18]. 

However, this approach requires an interactive communication style by care providers who have to be aware that people can prefer different styles of participation depending on age, health literacy [19], culture, and participation expectations [20]. 

In the Italian context, research on patient involvement in the caring process is in its infancy, and the existing studies highlight poor engagement in this process by healthcare professionals. Fontanesi and Goss [21] showed that practitioners gave low ratings to patient involvement, did not illustrate to patients all the alternatives to treat their problems, and did not ask patients about their preferences in receiving information. Most of the Italian literature includes articles on the importance of the topic [22] or reviews on the current situation [23,24], but there is no evidence on how healthcare professionals involve their patients in care processes and if there are differences between nurses and physicians. Indeed, most studies analyzed physicians’ and nurses’ approaches to SDM separately and focused on the healthcare professionals’ perspective rather than the point of view of the patient [25,26,27,28]. The only aspect that emerged concerned the area of patient information, highlighting the need for improved coordination and communication between nurses and physicians in order to provide information in a consistent and structured manner, thus reducing conflicting information for the patient [29,30,31]. 

To the best of our knowledge, other factors, such as patient satisfaction with shared decision-making and patient involvement in encounters with physicians and nurses, respectively, were not examined. Thus, the aim of this study was to investigate patients’ perceptions about shared decision-making and to compare patient ratings of their involvement in encounters with physicians and nurses so as to detect any differences. 

## 2. Materials and Methods

### 2.1. Design and Setting

The survey was a descriptive–observational type, and it included patients admitted to the general surgery and neurology departments of a university hospital in Italy. The primary outcome of the study was to examine patients’ perceptions about shared decision-making by administering a structured questionnaire to inpatients in the two wards and to compare and identify differences in patients’ assessment of their participation in encounters with physicians and nurses. In addition, as a secondary outcome, patient satisfaction with the quality of their participation in care and treatment was measured for physicians and nurses, respectively, and the relationship between the level of patient involvement, patient satisfaction, and perceived quality of care and treatment was determined. These measurements were performed at admission and at discharge to assess whether the communication interaction between professionals and patients changed during hospitalization. We expected higher patient ratings at discharge when professionals had adequate time to relate to patients and establish effective communication.

### 2.2. Participants and Recruitment

The study was carried out between March and June 2019, involving a convenience sample of patients from the general surgery and neurology departments of a tertiary care university hospital in southern Italy. Patients from the general surgery department were admitted for elective or emergency surgery such as thyroidectomy, mastectomy, herniectomy, lymph node surgery, breast cancer surgery, and laparoscopic abdominal wall surgery. The hospital includes about 220 beds plus about 50 beds in day hospital and day surgery. Patients from the neurology department were hospitalized for various health conditions such as amyotrophic lateral sclerosis, transient ischemic attack, stroke, migraine, epilepsy, and autoimmune neurological diseases. Participants’ inclusion criteria were being 18 years old or older, having been admitted to the ward for at least 48–72 h, and providing informed consent. The only exclusion criterion was the patients’ inability to understand the information and provide consent to participate in the study. Permission to conduct the study was obtained from ward managers. With the collaboration of the head nurse of the wards, patients who met the above criteria were contacted to participate in the study.

### 2.3. Instrument

The instrument used for data collection was a structured questionnaire consisting of both validated and ad hoc scales for the purposes of the research. The questionnaire included a demographic part (age, education, and time of hospitalization) and a part with measurement scales regarding patient involvement in shared decision-making, patient satisfaction, and perceived quality of treatment/care. The scales regarding both patient involvement in shared decision-making and patient satisfaction were split into two separate sections, one referring to physicians and the other referring to nurses. The items of the two sections were slightly adapted to the specific roles so that patients could report their perceptions while thinking about each professional category separately. The scale regarding the perception of the quality of treatment/care was inserted at the end of the questionnaire as a generic section. Patients had to answer by thinking about the overall quality of the received treatment and how he/she perceived him/herself with respect to their own self-care ability.

### 2.4. Patient Involvement in the Process of Shared Decision-Making (Split for Nurses and Physicians)

To examine this variable, the partnership-building communication scale by Arnetz et al. [5] was used. The scale includes a total of 16 items and consists of three factors: -Information (5 items): information that patients received about their condition/disease and its progress, why and how specific interventions and nursing treatments were provided, possible pain or discomfort, and possible complications; -Patient needs (7 items): clarity and understanding of received information, opportunity to ask questions (during hospitalization and at discharge), being treated with respect, the responsiveness of professionals to patient requests, and information about medications/treatments;-Treatment planning (4 items): patient involvement in choices and goals regarding treatment/examinations, planning post-discharge actions, and motivating empowerment for patients’ future health. 

For each item, patients were asked to indicate the degree of involvement using a Likert scale ranging from 1 (No, not at all) to 4 (Yes, completely).

### 2.5. Patient Satisfaction (Split for Nurses and Physicians)

The scale includes 3 ad hoc items regarding the degree of patient satisfaction with treatment and care relationship with professionals during hospitalization. A sample item was, “The way the doctor/nurse and I communicate about my treatment/care satisfies me”. Patients had to indicate their satisfaction degree according to a Likert scale ranging from 1 (No, not at all) to 4 (Completely).

### 2.6. Perceived Quality of Treatment/Care (General Scale, Not Split for Professional Category)

To analyze this variable, we used the scale by Arnetz et al. [15]. The scale included 5 items regarding the perceived quality of treatment/care (e.g., clarity of treatment/care goals, appropriateness, effectiveness, ability to manage the disease autonomously, and patient’s involvement in influencing the effectiveness of treatment/care). Patients rated these aspects using a Likert scale ranging from 1 (Poor) to 4 (Excellent).

### 2.7. Data Collection

Patients invited to participate were informed about the study aim, modality of questionnaire administration, and timing. They were also reassured about the anonymity of the questionnaire. Once the study purpose was read and discussed, patients signed their consent to participate. To capture information about possible changes in interaction behavior with patients by professionals, data collection took place at two times: at the time of admission―at least 48–72 h after admission―and at the time of discharge, thus to examine potential changes in patients’ perceptions of involvement with professionals during hospitalization. Among patients admitted in an emergency, some of them were approached after the surgical intervention when they were clinically stable and able to interact. The patients admitted for elective surgery were approached at least 2 days after admission so that they had time to have a communicative exchange with healthcare providers. To ensure matching between the two times of the questionnaire and simultaneously preserve patient anonymity, a unique code originated directly by patients through written instructions provided by the researchers was used. Elderly patients over 70 years old with visual or motor impairments were asked to choose whether to complete the questionnaire autonomously or with the guidance of the researcher. In the latter case, the researcher maintained an impartial attitude with patients without making judgments so as not to influence their responses. The questionnaire administration took place in the inpatient rooms. The questionnaires, once completed, were given directly to the researcher, who placed them in a locked box. Later, patients who participated in the administration at admission were contacted at discharge to complete the questionnaire at discharge. At this stage, the contribution of the head nurse of the wards was crucial to providing the researcher with information about patient discharge and the most appropriate time to administer the second questionnaire.

### 2.8. Ethical Statement

The study was approved by the local Ethics Committee (PROT. PG/2018/8801). The research was authorized by managers of the participating units. Head nurses of the wards were involved in the study to facilitate patient recruitment.

### 2.9. Data Analysis

Data analysis was performed through SPSS 20.0 Statistical Software (IBM, Armonk, NY, USA). Scale reliability was examined by using Cronbach’s Alpha coefficient. Descriptive analyses were carried out to analyze participants’ characteristics and the study variables. A t-test was performed to compare patients’ responses from admission to discharge in order to reveal potential changes between the two times. The Wilcoxon test was used to examine any differences in patients’ perceptions about their involvement in shared decision-making with physicians and nurses. Correlation and binary logistic regression analyses were conducted to examine the relationship between shared decision-making variables and both patient satisfaction and perceived quality of care/treatment. For logistic regression, cut-offs for high and low values of the study variables were defined based on sample distribution in percentiles, namely considering the individuals who were distributed below and above the 50° percentile. “High” values for the variables were used as a referral for the regression analysis. Odds ratios (ORs) and 95% confidence intervals (CI) were reported. 

The Wilcoxon test, correlations, and logistic regressions were carried out using the study variables at discharge.

## 3. Results

A total of 301 out of 319 hospitalized patients (94% response rate) participated in the study at admission. Among these, 151 patients also participated in the data collection at discharge (50.2%). Thus, the final sample used for the aim of this study included 151 patients who answered the two questionnaires at both admission and discharge.

### 3.1. Demographics 

The majority of the sample included male patients (*n* = 79, 52.3%). Regarding the patients’ age, 21.9% (*n* = 33) of them were aged 46–55 years, followed by those aged 56–65 years (19.9%, *n* = 30), and 66–75 years (17.9%, *n* = 27). The lowest percentages were for patients aged 36–45 years (13.9%, *n* = 21), 26–35 years (13.2%, *n* = 20), patients older than 75 years (7.9%, *n* = 12), and younger than or equal to 25 years (5.3%, *n* = 8). Regarding education, 35.8% of patients had a high school diploma, 32.5% had a middle school diploma, 19.3% had a bachelor’s degree or master’s degree, and 12.6% had an elementary school diploma. With regard to hospital stay, 39.1% of patients were hospitalized for 5 to 7 days, 35.1% of patients for 4 days, and 25.8% were hospitalized for more than 7 days (up to 15 days).

### 3.2. Perceived Shared Decision-Making Process, Patient Satisfaction, and Quality of Care/Treatment

Comparing patients’ responses between admission and discharge about involvement behavior (e.g., information, patient needs, treatment planning) by physicians and nurses, it was found that scores referring to physicians did not significantly change between the two times (*p* > 0.05), while patient satisfaction with the relationship with physicians increased significantly from admission to discharge (*p* < 0.05). Regarding nurses, for all the study variables, patient scores at admission increased significantly at discharge (*p* < 0.05). 

The general quality of treatment/care (e.g., appropriateness, effectiveness, and ability to manage the disease autonomously) perceived by patients did not significantly change from admission to discharge, even if a marginal improvement in the score was observed (Table 1).

However, shared decision-making scores referring to nurses were significantly lower than those referring to physicians for both the study times, while patient satisfaction with their relationship with both physicians and nurses did not differ at either time point (Table 2).

Differences in patients’ perception of shared decision-making were also observed with the Wilcoxon test. Specifically, 71 out of 151 patients (47%) rated the information received by nurses lower than that provided by physicians. A total of 16 patients (10.6%) answered that nurses gave more information compared to physicians, and 64 patients (42.4%) felt that nurses and physicians provided the necessary information to the same extent (Wilcoxon test: Z = −6.470, *p* < 0.001). Regarding patient needs, 55 patients (36.4%) reported that nurses cared about patients’ information needs to a lesser extent than physicians. However, 81 patients (53.6%) showed tie scores for the two professionals (Wilcoxon test: Z = −5.642, *p* < 0.001). For treatment/care planning, 41 patients (27.1%) reported being more involved in this process by physicians than nurses. Nevertheless, 83 patients (55%) reported that nurses and physicians involved patients in treatment/care planning to the same extent (Wilcoxon test: Z = −3.119, *p* < 0.01). Finally, the results showed that patients’ perceptions of satisfaction with the relationship with physicians and nurses are equivalent; 23 patients (15.2%) report higher satisfaction with the relationship with nurses and 21 patients (13.9%) with the relationship with physicians. Most patients (*n* = 117, 77.5%) report equal scores for both professionals. (Wilcoxon test: Z = −0.796, *p* > 0.05).

### 3.3. Relationship between Shared Decision-Making and Both Patient Satisfaction and Perceived Quality of Treatment/Care

Table 3 shows correlation analysis and Cronbach’s Apha (α) of the study variables. Correlations were in the expected direction. All the shared decision-making variables referring to both nurses and physicians were significantly and positively correlated to patient satisfaction and quality of treatment/care. The only non-significant correlation was between information provided by nurses and the perceived quality of treatment/care (r = 0.111, *p* > 0.05). The reliability indices of the used scales were all above the acceptability value (α ≥ 0.70).

Regression analyses regarding the relationship with nurses showed that higher levels of both patient satisfaction and perceived quality of treatment/care were significantly related to higher ratings of nurses’ attention to patients’ needs. Specifically, paying attention to patients’ needs increased the likelihood that patients were satisfied with the relationship experienced with nurses by almost four times and with the received treatment/care. The other aspects of shared decision-making, such as receiving information and involvement in planning treatment/care, were not significantly associated with the studied dependent variables. 

With regard to the relationship with physicians, the results showed that both higher patient satisfaction and quality of treatment/care were related to the fulfillment of patients’ information needs and to receiving relevant information. Specifically, patients who felt that their information needs were fulfilled were about three times more likely to be satisfied with their relationship with physicians and perceived higher quality of treatment/care. Similarly, patients who stated that important information about their health condition, exams, treatment, and health consequences were provided were about twenty times more likely to be satisfied with physicians and about eight times more likely to perceive a high quality of treatment/care. Patient involvement in planning treatment was significantly related to the perceived quality of treatment but not to patient satisfaction with physicians (Table 4).

## 4. Discussion

The aim of this study was to investigate patients’ perceptions of their involvement in the shared decision-making process with physicians and nurses, respectively. We also measured the level of patient satisfaction with the quality of involvement in caring/treatment and the relationship between the extent of patient involvement and both patient satisfaction and perceived quality of care/treatment. 

### 4.1. Patients’ Perceptions of Involvement in Shared Decision-Making Process

The results show that patients’ responses about shared decision-making variables (e.g., information, patient needs, treatment planning) were significantly different for physicians and nurses. Physicians seem to be more careful than nurses in involving patients in the process and seem to be more consistent in their behavior, as there were no significant differences in scores from admission to discharge. However, although nurses devoted less attention to involving patients and making them participants in the caring process, patients gave them significantly higher ratings at the end of their hospitalization, as shown by the significant increase in scores from admission to discharge. This may be due to the fact that nurses, because of their professional nature, spend more time with patients during the course of their hospitalization, and it can be the reason why scores for physicians remained stable over time. Moreover, a nurse is seen as a mediator between the physician and the patient. The nurse’s role in the shared decision-making process is not only to provide information but also to engage the patient. Interventions that facilitate this interaction include decision support tools and decision coaching. Some authors revealed that nursing staff could help patients understand their disease, clinical progress, and treatment options by using SDM. Indeed, patient decision support provides fact-based information about options, outcomes, and probabilities as a supplement to physician consultation, helps people determine the desirability of potential benefits versus potential harms, and guides patients through the decision-making process. Decision coaching is effective in helping patients consider informed values and in guiding decision-making. In this way, nurses can work with patients and equip them with the knowledge and tools they need to make conscious and autonomous decisions about their own health [27,32].

An aspect to consider is that the final decision maker in SDM is the physician, and culturally nurses are often subordinate to the physician [33]. This may explain the results that emerged from our study that show greater attention by physicians than nurses in involving patients in the caring process. However, nurses scored significantly higher on discharge than on admission, which is consistent with findings in the scientific literature. Indeed, nurses are a large and important member of the professional health care team; their involvement in the SDM process and their understanding of basic concepts and principles related to the decision-making process are particularly important [28,34]. They have a role as a health educator, advocate, data collector, symptom and side-effect responder, information sharer, and psychological supporter. [35] On the contrary, physicians typically do not have enough time to provide adequate information and support patients when they make decisions [29,36]. Nurses, instead, for their professional nature, may have more time to conduct SDM at the patient’s bedside.

### 4.2. Patient Satisfaction with Quality of Involvement in Caring/Treatment

Significant differences were found between physicians and nurses with regard to receiving relevant information about care/treatment, fulfilling patient needs, and involving patients in planning treatment/care. Specifically, most patients reported having received information from doctors about their health condition, treatments, and health consequences, partially or completely. On the contrary, nurses provided little or no information about nursing care needs, reasons, and modalities for nursing interventions. 

Similar findings are evident in relation to both the ability to meet patients’ information needs and their involvement in planning treatment/care, where scores were consistently higher in regard to physicians while concerning nurses, most patients report being little or not particularly involved in the shared decision-making process. Such differences could be due to several reasons. The medical doctor’s responsibility of informed consent to legitimize health treatment (Law No. 219, 2017) “obligates” physicians to relate to patients by providing them with all the necessary information about treatment, goals, and possible therapeutic options that may be implemented [37]. This formal act could be interpreted by patients as special attention from the physician. Regarding nurses, such a duty is not expected from their professional profile (Ministerial Decree No. 739, 1994); thus, the initiative to involve patients in caring is often left to their own professional ethics.

### 4.3. Relationship between the Extent of Patient Involvement and Both Patient Satisfaction and Perceived Quality of Care/Treatment

It is interesting to note that patients in this study reported being rather satisfied with physicians and nurses to the same extent. Among the factors determining patient satisfaction with nurses, meeting patients’ needs, such as the opportunity to ask questions, clarity of information, sensitivity to patients’ requests, and patient respect, was the only significant variable. The same result is found regarding the perceived quality of care for which nurses’ attention to patient needs appears to be crucial. This can probably be explained by the fact that, during hospitalization, nurses have more opportunities than physicians to build a relationship with patients, and they feel free and safe to ask questions and express their needs [38,39]. A literature review [40] reports some factors that contribute to increasing trust relationships between nurses and patients. Characteristics such as honesty, sensitivity, giving confidence to patients, striving to provide them with the best possible care, and, most importantly, placing trust in the patients’ ability to understand information and make decisions regarding their clinical condition facilitate patients’ trust in nurses. Both receiving information about caring and its modality and involving patients in planning care does not appear to affect either satisfaction with nurses or perception of quality of care. This result can be due to the fact that the two variables refer to behaviors that are passive for patients, while patient needs are proactively shown by them; thus, when they are listened to, they feel more satisfied with nurses and the quality of care.

Regarding both patient satisfaction with physicians and perceived quality of treatment, the most significant variable was receiving information during hospitalization. In fact, patients who perceived that physicians provided relevant information about their health, modality of treatments, and possible pain or general discomfort were about twenty times more likely to be satisfied with the relationship with physicians and about eight times more likely to perceive a high quality of treatment. Attention to patient needs is also a significant variable associated with satisfaction with physicians and perceived quality of treatment, although less so than receiving information. This finding supports what was said above about the medical responsibility of informed consent that makes physicians provide relevant information about health conditions and treatment to patients. Finally, patient involvement in planning treatment is not related to patient satisfaction, but it is related to the perception of the quality of treatment, probably because patients feel more proactive and can make decisions shared with the physician based on their specific needs [41,42].

### 4.4. Limitation

This study has a few limitations. First, although a good sample size was obtained at admission, at discharge, there was an important dropout of participants. This was mainly due to organizational problems because researchers were not always present on the ward at the time of discharge of the patients recruited at admission. In this sense, the final sample size of the study was poor, and it may not be adequately representative of the patients hospitalized during the study. Second, the study is monocentric, and the collected data refer to two units available to participate (convenience sample) from an Italian hospital. In order to ensure the generalizability of the results and greater external consistency, it would be desirable to involve units from other hospitals in future studies. Third, another limitation includes the modality of data collection. To make it easier for elderly patients with visual impairment to fill out the questionnaire, the researchers were supportive by guiding them during the completion. Although the researchers maintained an impartial attitude with patients without trying to influence their responses, this modality could result in a method bias by increasing social desirability [43]. Also, the instrument used was self-reported, which is useful for capturing perception data, but it was limited to the patient’s perspective only. To overcome this limitation, future research should involve multiple sources, such as the perspective of the professional (physician and nurse), or integrate along with patients’ perception, objective data such as compliance to the treatment and other patient outcome indicators such as health behaviors. However, a strength of the study was the longitudinal-type survey that allowed for examining the change in participants’ perception during hospitalization and, indirectly, the change in patient involvement behavior of professionals. Finally, the second survey at discharge allowed for collecting data when patients were hospitalized for at least four days of hospitalization, contributing to greater reliability of participants’ perceptions about the quality of communication with professionals and shared decision-making.

### 4.5. Practical Implications and Future Directions

This study suggests that it is important to sensitize professionals to involve patients in the process of care and treatment and provide relevant information about caring and treatment planning. This is supported by Ringdal et al. [44], who show that patients want to be active participants in the process, to know their health conditions, to share information with health professionals, and to be involved in decision-making. Future studies could analyze patients’ knowledge needs from a qualitative point of view and gain a deeper understanding of which activities or process phases they would like to be mainly involved in and whether there are relevant differences between young and elderly patients. The results showed that the key determinant of satisfaction with nurses is the attention to patients’ needs about questions concerning their care, clarity of received information, and sensitivity to compliance with requests/needs. On the other hand, providing appropriate information about disease progression and ways of performing examinations and treatments is a key determinant of satisfaction with physicians, who are perceived by participants as professionals who involve patients in the shared decision-making process more than nurses. Instead, no association emerges between patient involvement in care planning and either satisfaction with nurses or perceived quality of treatment. This would allow one to think that in the context studied, nurses rarely implement care planning by setting goals to be achieved or do not involve patients in goal identification or in care pathways necessary to achieve them. This suggests that at the local level, the intellectual dimension and autonomy of the nursing profession are not fully recognized yet, thus reinforcing the concept of the nurse as a subordinate and executor of physicians’ directives. For some years now, the nursing scientific community has been placing the acquisition of specialized skills for nurses at the core of the debate on the development of the profession. One example of that is the study by Saiani et al., in which the authors seek clarity on the development and recognition of advanced nursing skills [45].

However, to sensitize nurses to involve patients in the decision-making process, interventions such as training courses and simulations would be useful to prepare professionals to use effective communication techniques in different care situations [46], perhaps taking into account the age of patients. Moreover, to avoid patient involvement in decision-making becoming a source of stress for professionals due to an increased workload [47], it would be advisable to adequately plan work activities so that a portion of time in each work shift is devoted to communication with the patient [48].

Finally, a future direction of the study concerns the possibility of extending the research to nurses and physicians so as to understand their views and the factors hindering the shared decision-making process.

## 5. Conclusions

The results showed that physicians seemed to pay more attention to involving patients in the shared decision-making process than nurses did, although patients gave the later significantly higher ratings at the end of the hospitalization. However, an important result is that less than half of the patients considered that physicians and nurses provided the necessary information. In this sense, it is important that physicians and nurses foster an interprofessional SDM, including information sharing, discussion, and final treatment decision before discussing with the patient or family, so that the health providers can speak with a unified voice and meet patients’ information needs.

## Figures and Tables

**Table 1 ijerph-19-14229-t001:** Paired sample test comparing patients’ scores from admission to discharge.

	Admission (Mean; SD)	Discharge (Mean; SD)	Paired Differences					t	*p*
			Mean Dif.	SD	SE Mean	95% CI			
						LL	UL		
**SD-M Referring to Physicians**									
-Information	3.50; 0.81	3.50; 0.79	0.01	0.85	0.07	−0.13	0.14	0.09	0.92
-Patient needs	3.05; 1.05	3.17; 1.02	−0.12	0.88	0.07	−0.26	0.02	−1.67	0.10
-Treatment planning	2.76; 1.23	2.66; 1.23	0.09	1.08	0.09	−0.08	0.27	1.06	0.29
-Patient satisfaction	3.34; 0.89	3.47; 0.79	−0.13	0.66	0.05	−0.23	−0.02	−2.36	0.02
**SD-M referring to nurses**									
-Information	2.44; 1.23	2.62; 1.22	−0.17	0.89	0.07	−0.32	−0.03	−2.37	0.02
-Patient needs	2.27; 1.17	2.53; 1.24	−0.26	0.93	0.08	−0.41	−0.11	−3.40	0.00
-Treatment planning	2.12; 1.19	2.35; 1.21	−0.23	0.94	0.08	−0.38	−0.08	−3.03	0.00
-Patient satisfaction	3.41; 0.79	3.52; 0.69	−0.11	0.66	0.05	−0.21	−0.00	−1.99	0.05
Perceived quality of treatment/care (provider not specified)	3.29; 0.89	3.40; 0.84	−0.11	0.74	0.06	−0.23	0.01	−1.76	0.08

Note. SD-M = Shared decision-making, SD = standard deviation, SE = standard error, CI = confidence.

**Table 2 ijerph-19-14229-t002:** Paired sample test comparing patients’ scores with referring to nurses and physicians.

	Physician Mean; SD	Nurse Mean; SD	Paired Differences					t	*p*
			Mean Dif.	SD	SE Mean	95% CI			
						LL	UL		
**SD-M Perception at Admission**									
-Information	3.50; 0.81	2.44; 1.23	1.06	1.36	0.11	0.84	1.28	9.59	0.00
-Patient needs	3.05; 1.05	2.27; 1.17	0.78	1.29	0.11	0.57	0.98	7.37	0.00
-Treatment planning	2.76; 1.23	2.12; 1.19	0.64	1.22	0.10	0.44	0.83	6.41	0.00
-Patient satisfaction	3.34; 0.89	3.41; 0.79	−0.07	0.85	0.07	−0.20	0.07	−0.95	0.34
**SD-M Perception at discharge**									
-Information	3.50; 0.79	2.62; 1.22	0.881	1.42	0.12	0.65	1.11	7.60	0.00
-Patient needs	3.17; 1.02	2.53; 1.24	0.636	1.25	0.10	0.44	0.84	6.24	0.00
-Treatment planning	2.66; 1.23	2.35; 1.21	0.311	1.14	0.09	0.13	0.50	3.34	0.00
-Patient satisfaction	3.47; 0.79	3.52; 0.69	−0.046	0.67	0.06	−0.16	0.07	−0.82	0.41

Note. SD-M = Shared decision-making, SD = standard deviation, SE = standard error, CI = confidence interval of the difference, LL = lower limit, UL = upper limit.

**Table 3 ijerph-19-14229-t003:** Correlation analysis and Cronbach’s Alpha for the study variables.

**Variables**	**1. SD-M Referring to Physicians**				
1. SD-M referring to physicians	1a. Information	1b. Patient needs	1c. Treatment planning	2. Patient satisfaction with physicians	3. Perceived quality of treatment/care (provider not specified)
1a. Information	(0.86)				
1b. Patient needs	0.425 **	(0.82)			
1c. Treatment planning	0.344 **	0.420 **	(0.75)		
2. Patient satisfaction with physicians	0.723 **	0.423 **	0.424 **	(0.90)	
3. Perceived quality of treatment/care (provider not specified)	0.634 **	0.303 **	0.355 **	0.680 **	(0.89)
**Variables**	**1. SD-M referring to nurses**				
1. SD-M referring to nurses	1a. Information	1b. Patient needs	1c. Treatment planning	2. Patient satisfaction with nurses	3. Perceived quality of treatment/care (provider not specified)
1a. Information	(0.92)				
1b. Patient needs	0.740 **	(0.81)			
1c. Treatment planning	0.721 **	0.726 **	(0.91)		
2. Patient satisfaction with nurses	0.230 **	0.238 **	0.236 **	(0.86)	
3. Perceived quality of treatment/care (provider not specified)	0.111	0.161 *	0.215 **	0.596 **	(0.89)

Note. SD-M = Shared decision-making. * *p* < 0.05, ** *p* < 0.01. Cronbach’s Alpha values are shown in the diagonal in parenthesis.

**Table 4 ijerph-19-14229-t004:** Binary logistic regression results: relationship between shared decision-making variables and both patient satisfaction and perceived quality of treatment/care.

Variables in the Equation	Patient Satisfaction			Perceived Quality of Treatment/Care (Provider Not Specified)		
	OR	95% CI for OR		OR	95% CI for OR	
		LL	UL		LL	UL
SD-M referring to nurses						
Information	2.228	0.855	5.802	1.414	0.571	3.502
Patient needs	3.650 *	1.315	10.136	3.971 *	1.495	10.551
Treatment planning	0.945	0.433	2.062	0.842	0.389	1.825
SD-M referring to physicians						
Information	19.755 **	7.288	53.550	8.028 **	3.254	19.807
Patient needs	2.875 *	1.016	8.139	2.593 *	1.060	6.348
Treatment planning	2.850	0.902	9.000	2.779 *	1.065	7.250

Note. SD-M = Shared decision-making, OR = Odd ratio, CI = confidence interval, LL = lower limit, UL = upper limit. * *p* < 0.05, ** *p* < 0.001.

## Data Availability

Not applicable.

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
