# Peer review of "Patient Involvement in Shared Decision-Making: Do Patients Rate Physicians and Nurses Differently?"

_ijerph, 2022, doi:10.3390/ijerph192114229_

Round 1

Reviewer 1 Report

Please see file.

Author Response

Dear Editor, 
We would like to thank the reviewers for their thorough review work that helped to improve our paper. We have addressed all their requests and suggestions and we hope the changes made will meet their expectations.
You can find the reply to the reviewers' comments in the attachment.
Thank you 
Maura Galletta

Reviewer 2 Report

The authors took upon themselves an important task - of examining SDM in Italy, no doubt with the purpose of enhancing SDM there.

I wonder about the measures they used for SDM, and especially about using them for doctors and nurses alike. How can a nurse involve a patient in the treatment plan, when she does not determine it? She can inform, but not involve.  This makes the conclusion in the discussion and abstract regarding nurses somewhat questionable.  

It is also clear that patients rate nurses higher than doctors on most measures. Some of it perhaps because of what I related above. Some also because of culturally deferring to physicians. Can we hear something about it? I propose citing  the chapter on doctor-patient relationships  from Miron-Shatz, T. (2021). Your Life Depends On It: What You Can Do to Make Better Choices about Your Health. Basic Books.

An interesting issue emerged in the introduction when the authors wrote about previous studies that took place in Italy: "The only aspect that has emerged concerned the area of patient information, by highlighting the need for improved coordination and communication between nurses and physicians in order to provide information in a consistent manner that does  not mislead the patient [29-31].  Why mislead? Is this a cultural issue, as in not to worry the patient? Please add a sentence here.

Minor issues:

The authors report submitting the questionnaire to patients at admission. But then they write (lines 188-9) "at the time of admission―at least 48-72 hours after admission". Is it - at least, or - no more than?  And this needs to be clarified from the beginning.  This is especially crucial when remembering that some of the surgery patients came for emergency care.

Speaking of which - were there any differences between emergency and other patients in surgery? 

Finally, in the discussion section, the authors relate to the important issue of trust, but they did not test for it. Why?

The way of reporting the demographics is unique. I would expect to see an average and SD for age, and likewise for length of hospitalization.

Author Response

(The authors gave the same response as above.)

Reviewer 3 Report

Very interesting topic for potential readers of this Journal.

Some comments are made in favor of improving the current version of the manuscript.

Abstract: Methodology. Additional information could be succinctly added to the questionnaire given to patients.

Keywords: perhaps better to use the plural form for “physicians” and “nurses”.

Introduction: It is long. In this section alone, 31 references(72%) of the 43 citations provided are delimited. Perhaps part of the introduction would fit more into the discussion section.

Methodology. Additional succinct information on the content of the patient questionnaire is missing. Doubts arise, why were the General Surgery and Neurology services selected and not other Services or several Services of this Center? Was the sample size calculated? General characteristics of the hospital? Second or third level hospital? Number of beds, location (north or south of the country), care area?

Results. Some results are missing. Are there differences according to the service in which the patient was admitted (Neurology vs General Surgery? Are there differences according to age, cultural level, average length of stay?

Discussion. Part of this section are really results that should be included in the section for this purpose. More confrontation of the results obtained with other works related to the subject is lacking. It should be noted that in this contribution only 6 bibliographic citations are mentioned.

Bibliography. Only 15 references (35%) of the 43 provided are recent (equal to or less than 5 years). I would be grateful to add any additional citations.

Conclusions. This section is long. It does not fit the objectives of the study.
Perhaps a noteworthy result is that less than half of the patients (64 patients, 42.4% of the sample) considered that the doctors and nurses provided the necessary information.

Author Response

(The authors gave the same response as above.)

Round 2

Reviewer 3 Report

.- Thank the authors for their excellent work and great responsiveness (in time and quality).
.- The authors reply letter is excellent. .- The changes suggested by the team of reviewers are perceived very positively. So its possible publication could be valued.